# Reconstructing tumor evolutionary histories and clone trees in polynomial-time with SubMARine

**Linda K. Sundermann**[1,2,3], **Jeff Wintersinger**[1,2,3,4], **Gunnar Rätsch**[5,6], **Jens Stoye**[7], **Quaid Morris**[1,2,3,4,8] *

**1** Donnelly Centre for Cellular and Biomolecular Research, University of Toronto, Toronto, Ontario, Canada, **2** Vector Institute for Artificial Intelligence, Toronto, Ontario, Canada, **3** Ontario Institute for Cancer Research, Toronto, Ontario, Canada, **4** Department of Computer Science, University of Toronto, Toronto, Ontario, Canada, **5** Department of Computer Science, ETH Zurich, Zurich, Zurich, Switzerland, **6** Biomedical Informatics, University Hospital Zurich, Zurich, Zurich, Switzerland, **7** Faculty of Technology and Center for Biotechnology (CeBiTec), Bielefeld University, Bielefeld, North Rhine-Westphalia, Germany, **8** Computational and Systems Biology, Memorial Sloan Kettering Cancer Center, New York City, New York, United States of America

* MorrisQ@mskcc.org

**Data Availability Statement:** The authors confirm that all data underlying the findings are fully available without restriction. All simulated files are

## Abstract

Tumors contain multiple subpopulations of genetically distinct cancer cells. Reconstructing their evolutionary history can improve our understanding of how cancers develop and respond to treatment. Subclonal reconstruction methods cluster mutations into groups that co-occur within the same subpopulations, estimate the frequency of cells belonging to each subpopulation, and infer the ancestral relationships among the subpopulations by constructing a clone tree. However, often multiple clone trees are consistent with the data and current methods do not efficiently capture this uncertainty; nor can these methods scale to clone trees with a large number of subclonal populations.

Here, we formalize the notion of a partially-defined clone tree (*partial clone tree* for short) that defines a subset of the pairwise ancestral relationships in a clone tree, thereby implicitly representing the set of all clone trees that have these defined pairwise relationships. Also, we introduce a special partial clone tree, the *Maximally-Constrained Ancestral Reconstruction* (MAR), which summarizes all clone trees fitting the input data equally well. Finally, we extend commonly used clone tree validity conditions to apply to partial clone trees and describe SubMARine, a polynomial-time algorithm producing the *subMAR*, which approximates the MAR and guarantees that its defined relationships are a subset of those present in the MAR. We also extend SubMARine to work with subclonal copy number aberrations and define equivalence constraints for this purpose. Further, we extend SubMARine to permit noise in the estimates of the subclonal frequencies while retaining its validity conditions and guarantees. In contrast to other clone tree reconstruction methods, SubMARine runs in time and space that scale polynomially in the number of subclones.

We show through extensive noise-free simulation, a large lung cancer dataset and a prostate cancer dataset that the subMAR equals the MAR in all cases where only a single clone tree exists and that it is a perfect match to the MAR in most of the other cases.

available for download at: https://github.com/
morrislab/submarine_data. The numerical data
used in the figures are included in S1 Data.

**Funding:** LKS was partially funded by the
International DFG Research Training Group GRK
1906/1 (https://didy.uni-bielefeld.de/), and is now
funded by a MITACS elevate postdoctoral
fellowship (https://www.mitacs.ca/en/programs/
elevate#postdoc). Part of this work was performed
while LKS was affiliated with and funded by
Bielefeld University and ETH Zurich. JW is funded
by the NSERC CGS-D program (https://www.
nserc-crsng.gc.ca/Students-Etudiants/PG-CS/
CGSD-BESCD_eng.asp). GR is partly funded by the
"Swiss Molecular Pathology Breakthrough
Platform", funded by the ETH Special Focus Area
"Personalized Health Related Technologies", grant
number #106. QM is supported by an NIH grant
(P30-CA008748), an Associate Investigator Award
from the Ontario Institute of Cancer Research
(which partially supports LKS, https://oicr.on.ca/
investigator-awards/), a subgrant from the
Canadian Centre for Computational Genomics
technology platform funded by Genome Canada,
and holds a Canada CIFAR AI chair (https://cifar.ca/
ai/canada-cifar-ai-chairs/). The funders had no role
in study design, data collection and analysis,
decision to publish, or preparation of the
manuscript.

**Competing interests:** The authors have declared
that no competing interests exist.

Notably, SubMARine runs in less than 70 seconds on a single thread with less than one Gb of memory on all datasets presented in this paper, including ones with 50 nodes in a clone tree. On the real-world data, SubMARine almost perfectly recovers the previously reported trees and identifies minor errors made in the expert-driven reconstructions of those trees.

The freely-available open-source code implementing SubMARine can be downloaded at https://github.com/morrislab/submarine.

## Author summary

Cancer cells accumulate mutations over time and consist of genetically distinct subpopulations. Their evolutionary history (as represented by tumor phylogenies) can be inferred from bulk cancer genome sequencing data. Current tumor phylogeny reconstruction methods have two main issues: they are slow, and they do not efficiently represent uncertainty in the reconstruction.

To address these issues, we developed SubMARine, a fast algorithm that summarizes all valid phylogenies in an intuitive format. SubMARine solved all reconstruction problems in this manuscript in less than 70 seconds, orders of magnitude faster than other methods. These reconstruction problems included those with up to 50 subclones; problems that are too large for other algorithms to even attempt. SubMARine achieves these result because, unlike other algorithms, it performs its reconstruction by identifying an upper-bound on the solution set of trees and the amount of noise in the estimates of the subclonal frequencies. In the vast majority of cases we checked, i. e. an extensive noise-free simulation, a lung cancer and a prostate cancer dataset, this upper bound is tight: when only a single solution exists, SubMARine converges to it every time. When multiple solutions exist, our algorithm correctly recovers the uncertain relationships in 71% of cases.

In addition to solving these two major challenges, we introduce some useful new concepts for and open research problems in the field of tumor phylogeny reconstruction. Specifically, we formalize the concept of a partial clone tree which provides a set of constraints on the solution set of clone trees; and provide a complete set of conditions under which a partial clone tree is valid. These conditions guarantee that all trees in the solution set satisfy the constraints implied by the partial clone tree.

This is a *PLOS Computational Biology* Methods paper.

## Introduction

Tumors contain multiple major subpopulations of genetically distinct cancer cells [1, 2]. The evolutionary history of a cancer can be reconstructed using the allelic frequencies of the clonal and subclonal mutations in one or more bulk samples of a single cancer. Multiple samples from the same individual's cancer can be either spatially distinct [3] or longitudinal [4, 5]. *Clonal* mutations are present in all profiled cancer cells and were inherited from their most recent common ancestor; *subclonal* mutations are those that are present only in some, or one, of the subpopulations. Subclonal reconstruction algorithms infer the ancestral relationships among the subpopulations by constructing a *clone tree*; the genotypes of individual subpopulations can then be determined using this tree. These trees contribute to a better understanding

of cancer development and response to treatment [6, 7] by helping to identify key steps in cancer progression [8, 9].

Clone trees are directed, rooted trees whose nodes correspond to different subclones, where directed edges link parental subclones to their direct descendants. A *subclone* is a group of cells descended from a single founder cell; and corresponds to a subtree (or clade) of the phylogeny of the cancerous subpopulations. Methods to construct clone trees assume that these cells all inherit the mutations present in the founder cells unless those mutations are removed from the cell through a copy number loss of its genomic locus. Subclones are associated with a set of subclone-defining mutations which are present in this founder cell but not in its parental subclone. The root of the tree, called the *germline*, represents the embryonic cell, which is the founder of all cancer cells (and all other cells in the body). In most, but not all cancers, there is a single cancerous subclone that is the ancestor of all the others; this special subclone is called the *clonal population* and it is associated with the cancer's clonal mutations.

Although there has been substantial progress in developing algorithms to build clone trees from bulk tumor samples [10–22]; two key challenges remain: scaling algorithms to clone trees with many subclones and efficiently capturing uncertainty in the clone trees. These challenges persist even when mutation allele frequency measurements are very precise. Here we address these two challenges. First, we assume perfect accuracy in the allele frequencies and derive an algorithm called SubMARine. It runs in polynomial-time and for an input set of subclonal frequencies, it summarizes an upper bound on the solution set of clone trees using a partial clone tree, a new data structure that defines the ancestral relationships between the pairs of subclones. Second, we introduce a *noise buffer* into SubMARine that permits noise in the subclonal frequencies. When SubMARine is unable to find a valid partial clone tree assuming precisely measured frequencies, it identifies a uniform noise buffer, which is the upper bound on subclonal frequency error necessary to allow in order to find a valid partial clone tree. Optionally, SubMARine can reduce this uniform upper bound in a subclone and sample-specific way.

**Contributions.** Here we introduce and formalize the notion of a partially-defined clone tree, or *partial clone tree* for short. This representation is a partial solution to a clone tree reconstruction problem that defines a subset of the pairwise ancestral relationships between the subclones, as well as a set of possible parents for each subclone. A partial clone tree is not a tree itself, but it implicitly defines a set of clone trees, i. e., all those trees that (i) are consistent with the ancestral relationships defined in the partial clone tree and (ii) select their parents from the possible parent set. The partial clone tree is thus a polynomial-space representation of a potentially exponentially-sized set of clone trees.

We also introduce a special partial clone tree: the *Maximally-Constrained Ancestral Reconstruction*, or MAR for short, which provides a complete summary of pairwise ancestral relationship constrained by the input data. Specifically, when multiple clone trees provide identically good fits to the mutation allele frequency data, the MAR captures all (and only) the pairwise ancestral relationships shared by this solution set of clone trees.

Additionally, we describe a polynomial-time algorithm, SubMARine, that produces the *subMAR*, which approximates the MAR. The ancestral relationships defined in the subMAR are guaranteed to be subset of those present in the MAR. Through extensive noise-free simulation and two large real-world datasets, we demonstrate that the subMAR almost always perfectly recovers the MAR. In particular, when the MAR represents a single clone tree solution, the subMAR matches it in all our experiments. On the real-world datasets, SubMARine also reproduces the clone trees found using expert-driven reconstructions or much slower algorithms and identifies minor errors made by the experts. SubMARine is designed not only for the basic clone tree reconstruction problems commonly addressed by other approaches, but also for

more complex problems that are less often considered. The basic problems include only simple somatic mutations (SSMs), which are single nucleotide variants (SNVs) and small insertions and deletions (indels), and clonal copy number aberrations (CNAs). The extended version of SubMARine also considers subclonal CNAs. Notably, SubMARine runs in less than 70 seconds on a single thread with less than one Gb of memory on all datasets presented in this paper, including ones with up to 50 subclones.

Although the standard noise-free version of SubMARine is immediately applicable to many real-world clone tree reconstruction problems without modification, we introduce a noise-buffered version of SubMARine. This version estimates a minimum deviation from the input frequencies required for a valid partial clone tree to exist. We describe strategies to use noise-buffered SubMARine to explore the space of clone trees with good fits to the input frequencies.

## Background

To define CNAs, the genome is divided into *segments*, with neighboring segments having different allele-specific average copy numbers in one or more samples. CNA reconstruction algorithms identify these segments and infer the average allele-specific copy numbers within them [23, 24]. However, fewer algorithms indicate the evolutionary relationship among the CNAs [10, 14, 22, 25]. SSMs are quantified experimentally by reporting their variant allele frequencies (VAFs) in each sample as estimated by short-read sequencing. These VAFs can be transformed into estimates of the cellular frequency of the SSMs by accounting for clonal CNAs in the sample influencing this transformation [26]. SSMs can be grouped into subclones based on these inferred cellular frequencies, thus estimating the associated subclonal frequencies in each sample [27–29]. With some modifications, similar algorithms can also be used to group CNAs into subclones [30–32]. The accuracy of the cellular frequency estimates, CNA reconstructions, and subclonal groupings depends heavily on the sequencing depth, degree of aneuploidy, and purity of the samples [33]. However, even under the best of conditions, when there is high accuracy in all of these, there remain substantial challenges in clone tree reconstruction.

S1 Fig shows a clone tree that solves a clone tree reconstruction problem by representing the ancestral relationships among the subclones. The solution to a **clone tree reconstruction problem** is a valid clone tree for the following input, which can be derived from a subclonal reconstruction problem: $K$ subclones (including the germline); their subclonal frequencies in each of $N$ samples, represented by the subclonal frequency matrix $\phi \in \mathbb{R}^{K \times N}$; $L$ CNAs assigned to segments, subclones and parental alleles; and $J$ SSMs assigned to segments and subclones.

A clone tree is **valid** if it satisfies the tree, the lost allele, and the sum constraints. The **tree constraint** simply requires the clone tree, thus the ancestral relationships, to be consistent with an arborescence (i. e., a directed tree whose edges all point away from the root) whose root is the germline. The **lost allele constraint**, which applies to both CNAs and SSMs, insists that mutations cannot occur on segments lost in an ancestral cell (see Section I in S1 Text for more details). Finally, because subclones represent subtrees (or clades) of phylogenies, the subclonal frequencies of a subclone must be larger than or equal to the sum of frequencies of its children in all samples, hence a **sum constraint** [11, 15] on the frequencies must hold in the clone tree:

$$\phi(k, n) \geq \sum_{k' \text{ is child of } k} \phi(k', n) \text{ for all } n \in \{0, 1, \dots, N-1\},$$

where $0 \leq \phi(k, n) \leq 1$ is the frequency of subclone $k$ in sample $n$.

The **basic clone tree reconstruction problem** considers only SSMs and clonal CNAs and, as such, only needs to consider $\phi$ when searching for valid clone trees. This problem was shown to be NP-complete [11]. The **extended clone tree reconstruction problem**, introduced

here, requires additional input, including an impact matrix $\mathcal{M}$. We introduce the extended problem in Section "Extended SubMARine: Clone tree reconstruction with subclonal CNAs".

**Previous work.** Often, multiple clone trees solve a clone tree reconstruction problem because the input data does not provide sufficient constraints to select a single solution [12, 15, 34]. The theoretical implications of this were first formally studied in [35, 36]. When there are multiple solutions, clone tree reconstruction algorithms invent other criteria to select a single solution [13, 21, 34] or they report a (hopefully) representative subset of the solution set [10, 14, 15, 18, 20]. Other methods simply enumerate all possible clone tree solutions [11, 12, 19]; however, because the solution space of clone trees grows exponentially with the number of subclones, these enumeration methods are limited to problems with a small number of subclones.

Given multiple clone trees as input, some methods identify a single [37] or multiple [38] representative consensus trees in order to capture topological features of the solution space. However, a single consensus tree cannot represent ambiguity in the data, and optimal selection of multiple consensus trees is NP-hard [38]. Furthermore, these methods already require the potentially exponentially-sized solution set of clone trees to be enumerated as input. In fact, already the problem of counting the number of valid solutions to the basic clone tree reconstruction problem is #P-complete [36].

## Partial clone trees

A partial clone tree defines some but, generally, not all of the pairwise ancestral relationships between subclones. A defined relationship either requires one of the subclones to be an ancestor of the other, or requires that the subclone *not* be an ancestor of the other. Thus, a partial clone tree can be represented with an *ancestry matrix* $Z \in \{1, 0, -1\}^{K \times K}$, where:

$$Z(k, k') = \begin{cases} 1 & \text{if subclone } k \text{ is an ancestor of subclone } k' \\ 0 & \text{if subclone } k \text{ is not an ancestor of subclone } k' \\ -1 & \text{if subclone } k \text{ is a possible ancestor of subclone } k' \text{ (aka } undefined) \end{cases}$$

A (full) clone tree *completes* a partial clone tree if its implied pairwise ancestral relationships are consistent with the defined (i. e. non-negative) entries in $Z$. A partial clone tree thus implicitly represents the set of clone trees that complete it. Hence, a partial clone tree can be used to solve the *Maximally-Constrained Ancestral Reconstruction Problem*:

**Problem 1 (Basic maximally-constrained ancestral reconstruction problem).** Given the subclonal frequency matrix $\phi$ of a basic clone tree reconstruction problem $t$, identify the pairwise ancestral relationships between subclones present in all valid clone trees.

The **basic maximally-constrained ancestral reconstruction (MAR)** is the unique partial clone tree that solves this problem by defining the maximal set of all of the ancestral relationships shared by the solution set of clone trees for $t$, and leaving undefined all relationships that vary within the solution set (Fig 1). Note, however, that this does not necessarily mean that all clone trees that complete the MAR are solutions of $t$; but often they are (S2 Fig). Note also that the partial clone trees produced by SubMARine also include a possible parent matrix $\tau$, which further constrains the space of completing clone trees (see Section "SubMARine: Approximating the MAR" and Section III.2 in S1 Text for more details); however, this matrix is not required in the definition of the MAR.

Partial clone trees generalize ancestry graphs (or evolutionary constraint networks) used by previous algorithms [11, 12, 19] as a starting point for enumerating all valid clone trees. An ancestry graph is a directed, acyclic graph (DAG), in which two subclones $k$ and $k'$ are

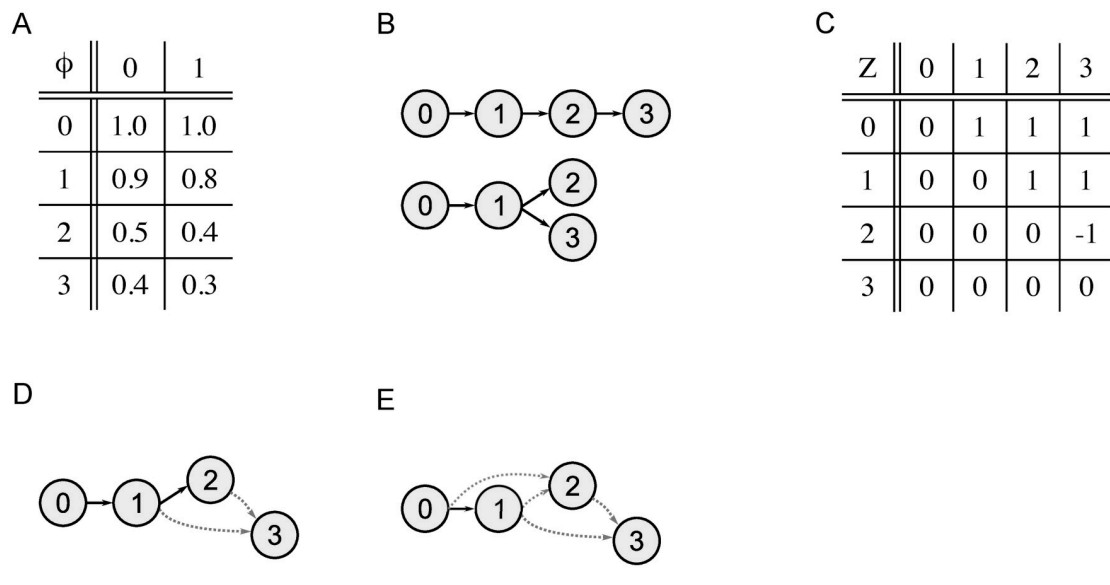

**Fig 1. Example of a MAR for a basic maximally-constrained ancestral reconstruction problem.** (A) The subclonal frequency matrix $\phi$ for the germline with index 0 and three subclones with indices 1-3 with their frequencies in two samples. (B) Set of valid clone trees that fit $\phi$. (C) The MAR summarizing the two clone trees, represented as ancestry matrix $Z$. Whenever subclone $k$ is an ancestor of subclone $k'$ in both clone trees of (B), $Z(k, k') = 1$. If $k$ is not an ancestor of $k'$ in both clones trees, $Z(k, k') = 0$. If $k$ is an ancestor of $k'$ in one clone tree but not in the other, as for subclones 2 and 3, $Z(k, k') = -1$. (D) The MAR drawn as a partial clone tree. Solid edges connect parents to their definite children (Eq (2)), dashed edges connect possible parents to their possible children (Definition 1 in Section III.5 in S1 Text). (E) A partial clone tree that does not equal the MAR. Here, subclone 1 is only a possible ancestor of subclone 2, although subclone 1 is the definite ancestor in both clone trees in (B). Hence, the defined set of ancestral relationships is not maximal.

connected by an edge if $k$ is a possible parent of $k'$. In these graphs, $k$ is a possible parent of $k'$ if there exists no sample $n$ such that $\phi(k, n) < \phi(k', n)$ (applying one aspect of the sum constraint) and if $k'$ does not contain any mutation that is already lost in $k$. Clone trees can be enumerated as spanning trees with a Gabow-Myers-based algorithm [39]; they are valid if the sum constraint is satisfied for each subclone and all its children. Ancestry graphs can be represented by a partial clone tree where $Z(k, k') = -1$ whenever an edge connects $k$ to $k'$, and where $Z(k, k') = 0$ otherwise. However, the semantics of a partial clone tree, which represents constraints on the ancestry, are not the same as the ones of an ancestry graph, which connects children to possible parents. Hence, not every ancestry matrix $Z$ with only 0 and $-1$ entries corresponds to an ancestry graph. Also, when a partial clone tree is represented as a DAG, not every spanning tree satisfying the sum constraint completes $Z$ (Section II.1 in S1 Text). Here, we extend this earlier work to include ancestry relationships that must be present (i. e., $Z(k, k') = 1$). Doing so allows us to not only more highly constrain the space of clone trees but also to propagate an initial set of defined ancestral relationships in $Z$ to infer other ancestral relationships that must appear in the MAR. We describe SubMARine, an algorithm that allows this propagation, in Section "SubMARine: Approximating the MAR".

## Applying validity constraints to partial clone trees

A key contribution of this paper is the observation that the validity constraints for clone trees can be applied to partial clone trees in order to rule out, or rule in, some pairwise ancestral relationships. In addition to the sum constraint, which is already applied in the construction of ancestry graphs, SubMARine enforces the tree constraint on $Z$. This allows to rule in certain ancestral relationships, i. e., identify pairs of subclones $k$ and $k'$ where $Z(k, k') = 1$. Doing so

permits us to define, for some subclones, a set of definite child subclones they have in every solution to a basic clone tree reconstruction problem $t$; which places further constraints on $Z$.

The tree constraint requires the clone tree to be an arborescence with the germline as the root. If we define clone 0 as representing the germline, we can immediately set $Z(0, k) = 1$ for $k > 0$ because the root is the ancestor of all nodes in the arborescence. This first consequence of the tree constraint is called the **germline constraint**. To simplify our presentation, we also assume that subclones 1 to $K − 1$ are sorted in decreasing order of their average subclonal frequencies across samples. As an obvious consequence of the sum constraint, this ensures that $Z(k, k') = 0$ whenever $k' \leq k$. Another consequence of the tree constraint arises from the fact that although all arborescences correspond to a unique, fully defined ancestry matrix $Z$; not all fully defined $Z$ matrices correspond to arborescences. To ensure a given $Z$ does represent such a tree, i. e., that it is transitive and each node has exactly one parent, it suffices to require that all the elements in $Z$ satisfy a single **partial tree constraint** (see Section II.2 in S1 Text for details):

$$Z(k, k') = Z(k, k'') \text{ if } Z(k', k'') = 1, \text{ for } k < k' < k''. \tag{1}$$

SubMARine can thus apply this constraint to partial clone trees to define an element of $Z$ whenever $Z(k', k'') = 1$ and either $Z(k, k') = −1$ or $Z(k, k'') = −1$ but not both.

To assist in applying the sum constraint to partial clone trees, we define a set of *definite* children of a subclone $k$. The definite children of a subclone $k, \chi(k)$, are the set of subclones whose parent can only be $k$ given the defined entries in $Z$:

$$\chi(k) = \{k' \mid Z(k, k') = 1\} \setminus \{k' \mid \exists\ k^{\circ} \text{ such that } Z(k, k^{\circ}) \neq 0 \text{ and } Z(k^{\circ}, k') \neq 0\}. \tag{2}$$

In other words, a subclone $k'$ is a definite child of subclone $k$ if $k$ is its ancestor, and $k'$ has no other (possible) ancestors that are (possible) descendants of $k$. (For Fig 1, the germline has only one definite child, which is subclone 1. Subclone 1 has subclone 2 as definite child, subclone 3 is a possible child of both subclones 1 and 2.) Thus, we can formulate the **generalized sum constraint** based simply on the set of definite children of a subclone:

$$\phi(k, n) \geq \sum_{k' \in \chi(k)} \phi(k', n) \text{ for all } n \in \{0, 1, \dots, N − 1\}. \tag{3}$$

Note that when there are no undefined states in $Z, \chi(k)$ is simply the set of all children of $k$. The **lost allele constraint** can be applied without any changes to a partial clone tree (Section II.3 in S1 Text).

Given these extended definitions of the validity constraints, we can now deem a partial clone tree to be *valid* if it satisfies the germline, generalized sum, lost allele, and partial tree constraints. We here note two things. First, the MAR is valid per construction (Section II.4 in S1 Text). Second, when $Z$ contains undefined states, some subclones have multiple possible parents and are not definite children of any subclone, hence these subclones are not considered in the generalized sum constraint. Thus, it is possible that a valid partial clone tree has no valid completions (S3 Fig).

## SubMARine: Approximating the MAR

SubMARine is a polynomial-time algorithm that constructs the *subMAR*, which is a partial clone tree that approximates the MAR. Here we describe the basic SubMARine algorithm, which approximates the solution to the basic maximally-constrained ancestral reconstruction problem. In the following section, we describe the extended version of SubMARine.

**Table 1. Overview of inference rules derived from the germline, generalized sum and partial tree constraint.**

| inference rule | functional description | impact | application |
|---|---|---|---|
| germline rule | $Z_0 \leftarrow f_{germ}(K)$ | $Z(0, k) = 0 \; \forall \; K > k > 0$ | once |
| generalized sum rules | | | |
| i) trivial relationships | $Z_0 \leftarrow f_{sum_{triv}}(K)$ | $Z(k', k) = 0 \; \forall$ $K > k' \geq k \geq 0$ | once |
| ii) crossing rule | $Z_0 \leftarrow f_{sum_{cr}}(\phi)$ | $Z(k, k') = 0$ (Eq (9) in Section III.1 in S1 Text) | once |
| iii) Subpoplar | $Z_{t+1}, \delta_{t+1}, \psi_{t+1}, \tau_{t+1} \leftarrow f_{sum_{subp}}(K, \phi, Z_t, \delta_t, \psi_t, \tau_t)$ | $Z(k, k') = 0$ if Eq (3) is violated when $k'$ was a child of $k$ and $k$ has no other (possible) descendants that are possible parents of $k'$, $Z(k, k') = 1$ if Eq (3) is not violated when $k'$ became a child of $k$ and $k'$ has no other possible parents than $k$ | once, and then every time a relationship is updated |
| partial tree rule | $Z_{t+1} \leftarrow f_{ptree}(Z_t)$ | $Z(k, k') = 1$ or $Z(k, k') = 0$ depending on two other defined relationships (Eq (1)) | once, and then every time a relationship is updated |

For explanation of available frequency $\delta$, definite parent vector $\psi$ and possible parent matrix $\tau$ see Section III.2 in S1 Text. $Z$ is the ancestry matrix, $\phi$ the subclonal frequency matrix and $K$ the number of subclones. Indices $k$ and $k'$ refer to subclones ordered by their average frequencies.

For a basic clone tree reconstruction problem $t$, the subMAR has three important properties, which we prove in this section: it is unique, its defined ancestral relationships are a subset of those in the MAR, and as such, all valid clone trees of $t$ are completions of the subMAR.

SubMARine takes as input the subclonal frequency matrix $\phi$ of a basic clone tree reconstruction problem $t$, and builds a partial clone tree by creating an ancestry matrix $Z$ (Algorithm 1 and S4 Fig). Initially, this matrix contains only undefined ancestral relationships. By applying inference rules derived from the validity constraints, SubMARine updates undefined values to defined ones whenever necessary, i. e. whenever undefined values violate constraints (Table 1). In a preprocessing phase, SubMARine applies the germline rule, setting $Z(0, k) = 1$ for all $k > 0$. Furthermore, all entries $Z(k', k)$, with $k' \geq k$, are set to 0 resulting from the sorting of subclones in decreasing order of their average subclonal frequencies across samples and as a consequence of the generalized sum constraint. Note that when multiple subclones have the same average subclonal frequencies, these are sorted according to user-provided IDs. Then, the main phase of the algorithm begins by applying the crossing rule that sets $Z(k, k') = 0$ for $k < k'$ whenever a sample $n$ exists such that $\phi(k, n) < \phi(k', n)$, as also required by the sum constraint. Afterwards, the last part of the generalized sum rule is propagated with our Subpoplar algorithm, which also propagates the partial tree constraint. This algorithm identifies definite children and rules out possible children. Its propagations lead to updates on $Z$ and on the set of possible and definite parents of each subclone, which is tracked in the possible parent matrix $\tau$. This tracking is necessary because the generalized sum rule can exclude possible parents for a subclone without requiring specific pairwise ancestral relationships (i. e., a subclone $k$ that cannot be a possible parent of subclone $k'$ can still be its possible ancestor). Whenever Subpoplar updates a relationship because of the generalized sum rule, the partial tree rule is propagated. When no more relationships can be defined through the inference rules, $Z$ converged and is output as the subMAR, together with the possible parent matrix $\tau$. Sections III.1 and III.2 in S1 Text provide more detailed descriptions of SubMARine and Subpoplar, along with an analysis of their polynomial runtime.

**Algorithm 1** Functional description of the SubMARine algorithm in basic mode.

```
Input: subclonal frequency matrix φ ∈ ℝ^{K×N}
Output: ancestry matrix Z, possible parent matrix τ
  ▷set 1's through germline constraint and 0's through trivial rela-
tionships of generalized sum constraint
```

```
1: Z_0 ← initializeCloneTree(K)
   ▷set 0's through crossing rule (Eq (9), Section III.1 in S1 Text) of
generalized sum constraint
2: Z_1 ← Z_0 ∪ f_{sum_{cr}}(φ)
   ▷set 1's and 0's through generalized sum rule with Subpoplar
algorithm
3: Z_2, τ_2 ← useSubpoplar(K, φ, Z_1)
4: return Z_2, τ_2
initializeCloneTree(K):
5: Z ← {−1}^{K×K} ∪ f_{germ}(K) ∪ f_{sum_{triv}}(K)
6: return Z
useSubpoplar(K, φ, Z):
7: initialize δ, ψ, τ
8: while Z did not converge do
9:      Z, δ, ψ, τ ← f_{sum_{subp}}(K, φ, Z, δ, ψ, τ)
10:     Z ← f_{ptree}(Z)
11: return Z, τ
```

Note that SubMARine always converges because only undefined values are updated to defined ones and their number is finite. At convergence, $Z$ represents a valid partial clone tree. If the subclonal frequency matrix $\phi$ does not support a valid partial clone tree—if, for example, one inference rule requires $Z(k, k') = 0$ and another requires $Z(k, k') = 1$, then SubMARine terminates and indicates the pair $(k, k')$ having a validity constraint violation. If the violation results from the generalized sum rule, this may be because the subclonal frequencies are not measured precisely but are actually inferred from noisy mutational frequencies. To address this issue, we describe a noise-buffered version of SubMARine in Section III.3 in S1 Text. In polynomial time, this version uses a binary search to find a minimum uniform noise buffer, which is a value that is added to the available frequencies of all parental subclones in order to permit a valid partial clone tree. Starting from the subMAR computed with this uniform buffer, SubMARine can also find a set of subclone- and sample-specific noise buffers and its corresponding subMAR, such that all completing clone trees use the lowest possible buffer. If the data allows, this set can be found in polynomial time. Otherwise, a depth-first search is necessary.

If the user decides to specify additional ancestral relationships for $Z$, they are added after the preprocessing phase, followed by a propagation of the partial tree rule (S4 Fig and Section III.1 in S1 Text). Furthermore, the partial tree rule is already propagated when applying the crossing rule. As additional input, clonal CNAs and SSMs can be provided. SubMARine checks then whether any SSMs are assigned to deleted segments and thus invalidate all clone trees through violating the lost allele constraint (Section III.4 in S1 Text). If this is not the case, the algorithm can proceed as previously described.

**Correctness.** As described previously, the inference rules used by SubMARine change only undefined ancestral relationships to defined ones and only when, given all of the other defined relationships, one of the two possible defined ancestral relationships causes a violation of the validity constraints. So, given a starting set of defined relationships associated with a clone tree reconstruction problem $t$, each relationship defined by one of SubMARine's inference rules is required in all valid clone trees that solve $t$. Thus, the subMAR's defined relationships are a subset of those in the MAR.

The constructed subMAR, given the subclonal frequency matrix $\phi$ of $t$, is unique because the order in which the inference rules get applied does not matter as long as all rules are applied and propagated until convergence. It is easy to show that the order of application is unimportant. Imagine a case where SubMARine generates two different subMARs, both starting from the same initial set of defined relationships, but that differ due to the order in which

the inference rules were applied. Because each subMAR's defined relationships are a subset of those in the MAR, so long as the MAR is defined (i.e., there is at least one valid and complete clone tree solution), all pairwise relationships that differ between these two subMARs are defined in one subMAR and undefined in the other. None of SubMARine's inference rules depend on an undefined relationship in order to update another undefined relationship. As such, there must be a path of inference rules linking all defined relationships shared by the two subMARs to each defined relationship unique to one of the two subMARs. Because this path exists, and the relationship is undefined in one of the two subMARs, the inference rules have not been propagated to convergence in the subMAR where the relationship is undefined. Ergo, so long as the inference rules are propagate to convergence, and the MAR is defined, two subMARs generated from the same starting point, using the same rules, are identical. As such, the subMAR is unique.

In summary, because (i) all ancestral relationships defined in the subMAR are a subset of those in the MAR and (ii) the subMAR is unique, all valid clone trees of *t* are completions of the subMAR.

SubMARine is implemented in Python and can be downloaded at https://github.com/morrislab/submarine. Next to the algorithm, we provide an implementation of a depth-first search to enumerate the set of valid subMAR-completing clone trees and an upper bound on the size of this set (see Section III.5 in S1 Text for a derivation of this bound).

## Extended SubMARine: Clone tree reconstruction with subclonal CNAs

The extended version of SubMARine propagates inference rules like the basic version but is designed specifically to include subclonal CNAs. For example, unlike the basic version, it propagates the lost allele rule; because whether or not the lost allele constraint is satisfied depends on the choice of clone tree. Its subMAR, which we call the *extended subMAR*, defines not only the set of valid clone trees but also a set of equivalent ones and approximates the *extended maximally-constrained ancestral reconstruction problem* defined below. Two clone trees are **equivalent** if they fit the experimental data equally well and if the same set of subclonal CNAs has the same impact on the mutant copy numbers of the same set of SSMs. Given subclonal frequencies and the assignment of SSMs and clonal CNAs to subclones, as in the basic version of SubMARine, the data fit does not depend on the ancestral relationships in the clone tree [20]. However, with subclonal CNAs, ancestral relationships can influence data fit. Specifically, subclonal CNAs change the VAFs of SSMs by altering their mutant copy numbers per cancer cell but only if 1) the subclonal CNA is in a descendant subclone, 2) the SSM is in the segment affected by the CNA and 3) the SSM is on the same parental allele, i. e., it has the same *phase*, as the CNA. As such, changing the ancestral relationship between an SSM-containing subclone and a CNA-containing one, can change the fit of the clone tree to the experimentally-measured VAF data. Note that because we model the change in CNA state, rather than the absolute copy number, the data fit to the experimental-derived average copy numbers of segments is not affected by the clone tree, see Section IV.1 in S1 Text for details. We represent the impact of CNAs on SSMs in an impact matrix $\mathcal{M} \in \{0, 1\}^{J \times L}$, where $J$ is the number of SSMs and $L$ the number of CNAs:

$$\mathcal{M}(j, l) = \begin{cases} 1 & \text{if the mutant copy number of SSM } j \text{ is changed by CNA } l, \\ 0 & \text{otherwise.} \end{cases}$$

As an aside, defining $\mathcal{M}$ requires us to assume each SSM is unique, i. e., we make an infinite sites assumption, otherwise we would not be able to select which version of the SSM is impacted by the CNA. Given the above, if two clone trees with the same subclonal frequencies and mutation assignments imply the same impact matrix, they also have equal data fit and are thus equivalent. Note that it is possible but exceptional rare, for two clone trees to have the same data fit but not the same impact matrix (see Section IV.2 in S1 Text for an example).

As indicated above, a CNA changes an SSM's mutant copy number only under specific conditions; thus the impact matrix $\mathcal{M}$ requires the presence and absence of specific ancestral relationships and SSM phases. These conditions, the **equivalence constraints**, are formally described in depth in Section IV.3 in S1 Text and their derived inference rules are propagated by extended SubMARine.

In the **extended clone tree reconstruction problem**, one is given a subclonal frequency matrix $\phi$; $L$ CNAs assigned to subclones, segments and parental alleles; $J$ SSMs assigned to segments and subclones; as well as an impact matrix $\mathcal{M}$; and is required to find a valid clone tree with subclonal CNA impacts that match $\mathcal{M}$. Given the input of an extended clone tree reconstruction problem $t$, the **extended maximally-constrained ancestral reconstruction problem** is to identify the pairwise ancestral relationships between subclones present in all valid clone trees that solve $t$ and are thus equivalent. The **extended MAR** is the unique partial clone tree that solves this problem by defining all, and only, the ancestral relationships as well as SSM phases shared by the solution set of valid and equivalent clone trees for $t$.

Like the basic subMAR, the **extended subMAR** has three important properties for an extended clone tree reconstruction problem $t$: its defined ancestral relationships and SSM phases are a subset of those in the extended MAR, it is unique, and consequently, all valid and equivalent clone trees of $t$ are completions of the extended subMAR (see end of Section IV.6 in S1 Text for more details).

As input, the extended version of SubMARine takes the subclonal frequency matrix $\phi$, CNAs as copy number changes (i. e., gains or losses) assigned to subclones, segments and parental alleles, SSMs assigned to segments and subclones, and the impact matrix $\mathcal{M}$ of an extended clone tree reconstruction problem (S5 Fig). Copy number changes, subclones, segments and alleles of the CNAs can be provided by subclonal CNA reconstruction methods [12, 14, 22, 25]. The impact matrix $\mathcal{M}$ can be easily derived from an existing subclonal reconstruction—then SubMARine generalizes from one clone tree to the set of valid and equivalent ones —but in some cases it can also be inferred without a subclonal reconstruction (Section "Conclusion and discussion"). For extended SubMARine the input CNAs have to satisfy a **monotonicity restriction**, which ensures that each segment contains only copy number changes of the same direction per allele (see Section IV.4 in S1 Text for details). In brief, this condition guarantees that once an allele is lost, no update of undefined ancestral relationships can prevent this loss from happening (e. g. by increasing the copy number of the allele before the loss), and hence no subsequent updates to $Z$ can remove conditions that used the lost allele rule to previously define an element of $Z$. This guarantees that all defined values in the subMAR set by propagating inference rules are present in the extended MAR. Note that copy neutral loss of heterozygosity (LOH) events can still be modeled because the restriction permits one of the parental alleles to be lost, and the other one to be gained.

Briefly, like the basic version of SubMARine, the extended version builds a partial clone tree by propagating the germline, generalized sum and partial tree rule and using the Subpoplar algorithm (S5 Fig). Furthermore, extended SubMARine propagates the equivalence and lost allele rules, and phases some SSMs in order to satisfy the underlying constraints (S1 Table and S1 Algorithm). In addition to user-defined ancestral relationships, the extended version of SubMARine can also take SSM phases as input. Extended SubMARine converges when no

ancestral relationship or SSM phase can be propagated anymore. As SubMARine in basic version, the extended version always converges. Its result is an extended subMAR, consisting of the ancestry matrix $Z$, the possible parent matrix $\tau$ and the SSM phasing $\pi_s$. An example of extended SubMARine and a detailed description of the algorithm, including an analysis of its polynomial runtime, can be found in Sections IV.5 and IV.6 in S1 Text.

## Results

Here, we evaluated SubMARine by applying it to simulated basic and extended clone tree reconstruction problems, thus without and with CNAs; and by applying it to data from the large, multi-sample TRACERx study [40, 41] and a CNA-containing prostate cancer study [42].

### Simulated data without noise

Section V.1 in S1 Text provides a detailed description of the creation of our noise-free simulated datasets. In brief, we generated a dataset without CNAs containing 600 subclonal reconstructions, evenly divided between those with 5, 20 and 50 subclones (plus the germline); and another dataset with clonal and subclonal CNAs containing 1800 subclonal reconstructions, each with 20 subclones. The CNA-containing subclonal reconstructions are evenly divided among nine groups of simulations where we try all combinations of the number of segments, selected from 10, 20, and 40, and the number of CNAs, selected from 10, 20, and 40. In each of the CNA-containing datasets, we randomly assigned CNAs as copy number changes to subclones, segments, and parental alleles, ensuring that a deletion is only allowed once per segment and allele on a given tree branch. We also randomly assigned 200 SSMs to subclones, segments, and parental alleles, considering the impact of subclonal CNAs. For both types of datasets and each parameter combination, we draw 10 random subclonal reconstructions for each of 1 to 20 samples, resulting in 200 subclonal reconstructions for each parameter combination.

SubMARine constructed each subMAR (basic or extended) in less than 70 seconds using a single thread with less than one GB of memory. On average, increasing the number of samples or decreasing the number of subclones decreases uncertainty in a clone tree [35, 36]. The implied ambiguity in the subMAR solutions shows the same behavior when applied to our simulations (S6, S7 and S8 Figs). Including CNAs in our simulations further decreases uncertainty (S6 and S8 Figs) due to the additional implied ancestral constraints. Notably, in all but one simulations with twelve and more samples, the resulting subMAR had no undefined ancestral relationships, indicating that it had found the single clone tree solution to the reconstruction problem.

We then assessed how accurately SubMARine's subMARs matched the actual ambiguity in the solution sets of clone trees fitting the 2400 clone tree reconstruction problems. Because each solution set is a subset of the clone trees completing the subMAR, we used a depth-first search (DFS) algorithm that incorporated the subMAR and the Subpoplar algorithm to enumerate these solution sets. Note that because not all spanning trees complete the subMAR (Section II.1 in S1 Text), we do not use the Gabow-Myers-based algorithm previously employed for this task [11, 12, 19]. For 1795 of the 2400 clone tree reconstruction problems, the subMARs were completely defined, so they only had a single clone tree solution. Among the remaining 605 problems, none of the problems predicted to have multiple solutions by SubMARine had only a single clone tree solution. So, in 100% of problems with a single solution, SubMARine identified that solution. Of the remaining 605 problems, in 64 cases, our DFS algorithm did not complete its enumeration in less than 120h on a single thread.

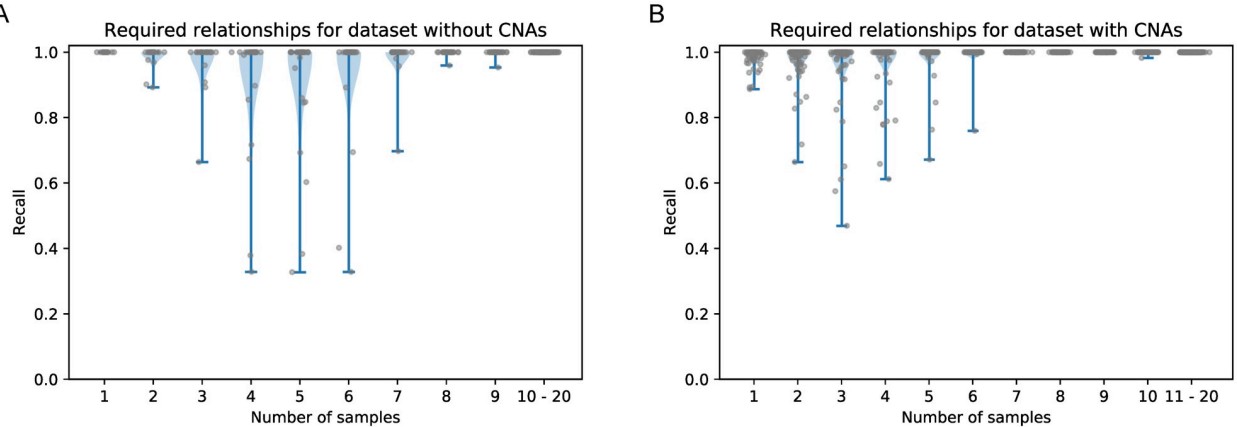

**Fig 2. Recall of defined ancestral relationships for dataset (A) without and (B) with CNAs.** We computed recall based on the non-trivial ancestral relationships. Columns in (A) usually have 30 data points, columns in (B) 90. The last column in each subfigure shows all results for (A) 10 and (B) 11 and more samples since each subMAR achieved a recall of 100%. For the 64 subMARs for which the DFS could not enumerate all valid (and equivalent) completing clone trees, we did not compute the recall because we do not know the ground truth. Hence, column 1 of (A) contains only 13, column 2 20, column 3 21 and column 4 27 values, and column 1 of (B) contains only 65 values.

For 70.9% of the 541 clone tree reconstruction problems for which we were able to fully enumerate the solution set, and that SubMARine predicts to have $> 1$ clone tree solution, the subMARs precisely matched the MAR. For all 2336 subMARs, for which we could compute their MAR, we computed the recall of defined relationships, i. e., the proportion of the non-trivial defined ancestral relationships (those for subclones $k$ and $k'$ where $0 < k < k'$) recovered from the MAR. Trivial ancestral relationships are those with which $Z$ is initialized in the pre-processing phase of SubMARine. As Fig 2 illustrates, the more constrained the clone tree reconstruction problem is by a higher number of samples, the higher is the recall. The presence of CNAs has no influence on the recall (S9 Fig). With CNAs, there is 100% recall with eleven or more samples, without CNAs, this is true for ten or more samples.

As Fig 3 illustrates, it may be possible to assess when the subMAR is a perfect match to the MAR. For the dataset without CNAs, all subMARs with 5 subclones have 100% recall (Fig 3A) as do the vast majority of subMARs with less than 50 undefined relationships (Fig 3B and 3C).

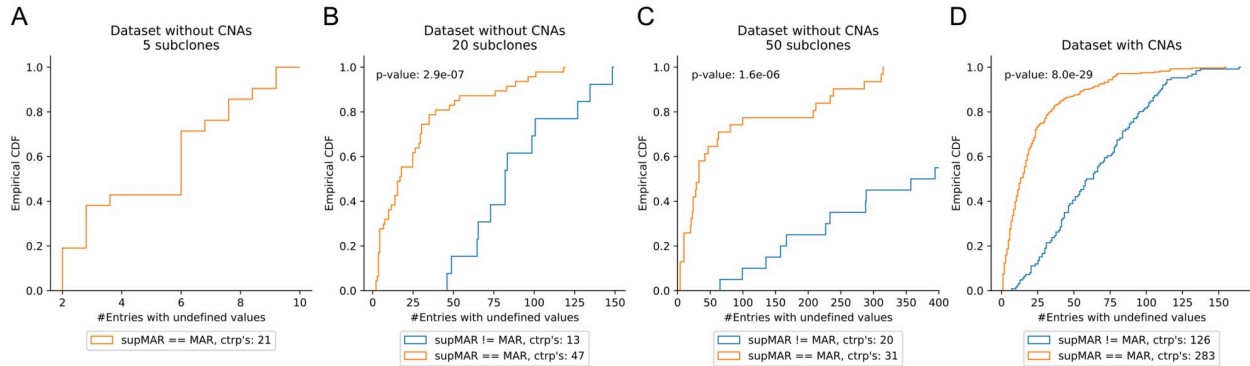

**Fig 3. Empirical cumulative density functions (CDFs) of subMARs equaling and differing the MAR for (A)–(C) dataset without CNAs and (A) 5, (B) 20 and (C) 50 subclones, and (D) dataset with CNAs and 20 subclones.** Not included are subMARs that do not contain any undefined ancestral relationships (and thus have found the single clone tree solution and equal the MAR), and those for which the DFS did not finish. The p-values are computed with a Kolmogorov-Smirnov test. (C) The empirical CDF for subMARs differing the MAR reaches the value of 1.0 at 864 undefined relationships. ctrp's: clone tree reconstruction problems.

For the dataset with CNAs, predicting when a subMAR has 100% recall is less straightforward as there is less than perfect recall with as few as 10 undefined relationships in the subMAR (Fig 3D). However, in the CNA-containing cases, the DFS is feasible to apply for subMARs with less than 50 undefined relationships as it was done in less than 100 seconds (S10 Fig).

## Simulated data with noise

Next, we assessed the performance of SubMARine on simulated data where we estimated the subclonal frequencies from read counts. As such, these frequencies contain an estimation error or noise. For performance assessment, we created a dataset containing 5400 subclonal reconstructions where count data was generated nine times from each of the 600 noise-free subclonal reconstructions with 5, 20, and 50 subclones and without CNAs from Section "Simulated data without noise". Each time, we used a different parameter combination of the number of SSMs per subclone and the total read count per SSM locus, resulting in seven different effective read depths (30, 100, 300, 1000, 3000, 10000, 30000), were two parameter combinations let to an effective read depth of 300 and two to 3000. For each subclone, its estimated frequency is based on variant read counts sampled from appropriate Binomial distributions (see Section V.2 in S1 Text for more details).

We applied SubMARine in basic mode with the option of using a noise buffer on the subclonal frequency matrices of all 5400 subclonal reconstructions. Since some of the subclonal reconstructions had multiple subclones with the same average subclonal frequencies across samples and SubMARine then simply sorts them according to their user-provided IDs, we provided SubMARine with subclonal IDs corresponding to the order of sorted subclones of the noise-free datasets. SubMARine built each subMAR in less than 35 seconds using a single thread with less than one GB of memory. Higher effective read depth provides more precise estimates for the subclonal frequencies, and as expected, S11 Fig shows that higher effective read depths increase the proportion of subMARs not requiring a noise buffer, and the proportion for which a set of subclone- and sample-specific noise buffers could be found directly without a depth-first search. With higher effective read depth, the maximum value in the noise buffer set becomes also smaller (S12 Fig). Note that for datasets with only one sample, a noise buffer is never necessary because a linear valid clone tree exists always.

In general, uncertainty in the subMAR also decreases with increasing effective read depth (S13 Fig), however, there are some key differences between the noisy simulations and the noise-free ones. In general, like in the noise-free case, increasing the number of samples decreases uncertainty, but how quickly this decrease occurs depends on the effective read depth and the number of subclones (S14, S15 and S16 Figs). However, in contrast to the noise-free case, when the effective read depth is below some cutoff that depends on the number of subclones, increasing the number of samples does not decrease uncertainty in the reconstructions. For example, for 50 subclones and an effective read depth of 100, adding more samples does not improve the reconstruction, while at the same effective read depth level for 5 subclones, no uncertainty in the reconstruction exists for eleven or more samples. Note that although subMAR uncertainty is only an upper bound on uncertainty in the MAR, it is possible that this is a general property of clone tree reconstruction problems.

To assess the accuracy of SubMARine's reconstructions on the noisy data, we compared each subMAR to the MAR of the corresponding noise-free subclonal reconstruction. Similar to the noise-free case, we computed the recall of defined ancestor-descendant relationships. However, because noise can change the ordering of the subclones, we considered more entries in the ancestry matrix $Z$ when computing the recall (Section V.3 in S1 Text). In addition, in the noisy case, it is possible that the subMAR contains defined ancestor-descendant

## Mean recall and percentage of error for effective read depth

**Fig 4. Mean recall and percentage of false positive errors for effective read depth.** With increasing effective read depth of 30, 100, 300, 1000, 3000, 10000, and 30000, the recall increases and the false positive errors of defining a relationship that is defined differently or undefined in the MAR decreases.

relationships not present in the noise-free MAR for the subclonal reconstruction, which creates false positive defined ancestor-descendant relationships. Here, we distinguish two types of false positives, where either the corresponding ancestor-descendant relationship in the noise-free MAR is undefined (deemed *falsely defined*) or where it is defined differently (deemed *differently defined*), and compute their percentage of error (Section V.3 in S1 Text). As before, we evaluate the performance of SubMARine only on those datasets for which we were able to build the MAR (Section "Simulated data without noise"). As shown in Fig 4, with increasing effective read depth, the recall increases while the two errors decrease. Note that the differently defined relationship error decreases more rapidly with increasing effective read depth than the falsely defined error (S17 Fig). By comparing recall, false positive errors, and sample-dependent reconstruction uncertainty with the noise-free cases (S14, S15, S16 and S18 Figs), we can define sufficient effective read depth levels for which noisy data is nearly equivalent to noise-free data. For 5 subclones and an effective read depth of 300, the mean recall does not differ from the one of the noise-free data, and the false positive error of differently defining a relationship in the subMAR is nearly 0. Furthermore, the proportions of subclones with uncertain parentage is not worse than in the noise-free dataset. Hence, an effective read depth of 300 for 5 subclones seems to be sufficient in order to construct a subMAR that represents the true underlying MAR well. For 20 subclones, an effective read depth of 10000 is preferable and for

50 subclones of 30000 or more. Note that effective read depth is the required read depth if each subclone is represented by only a single SSM; more SSMs correctly assigned to the subclones proportionally decrease the required read depth per SSM.

### TRACERx data

We next applied SubMARine to a large, multi-sample dataset drawn from the TRACERx study [41], consisting of mostly primary tumors of 100 patients with early-stage non-small-cell lung cancer (NSCLC). Previously, PyClone [28] was used for each patient to identify mutation clusters, which correspond to subclones, and CITUP [34] was used to infer clone trees by exhaustively exploring all possible trees and reporting those with the highest likelihood. In Section V.4 in S1 Text, we describe how we arrive at 88 patients with two to 15 subclones from two to seven tumor samples, on which we apply the basic version of SubMARine (S2 Table).

For each patient, SubMARine constructed the subMAR in less than 40 seconds on a single thread with less than one Gb of memory. For 42 patients, we did not use a noise buffer because their subclonal frequencies supported a valid partial clone tree; 37 of those have a subMAR that describes only a single tree. S19 Fig shows the five subMARs with undefined ancestral relationships. All five subMARs were identical to their MARs. In order to build a valid partial clone tree for the other 46 patients, we computed subclone- and sample-specific noise buffer sets (Section III.3 in S1 Text). For 45 of these patients, the noise buffer sets could be found in polynomial time. Only for one patient (CRUK0016), an exhaustive search had to be applied; it found the MAR and the noise buffer set in less than 2 seconds. The maximum values in the noise buffer sets range from 0.01 to 0.7 (S20 Fig), with a median of 0.14. Only one patient required a buffer greater than 0.5 (S21 Fig), this could be caused by infinite sites violations [43] or an undetected CNA. With the noise buffers, SubMARine identifies 42 additional subMARs that describe a single tree. For three of the four remaining patients, SubMARine finds subMARs with one, three and seven uncertain values being perfect matches to their MARs. The MAR of the remaining patient CRUK0016 contains nine undefined values.

We next compared SubMARine's partial clone trees with those clone trees reported in the TRACERx paper (p.31–p.174 of the Supplementary Appendix 1 of the work of Jamal-Hanjani et al. [41]). All but the trees for six patients were generated by CITUP. Full details of this comparison are provided in S2 Table. CITUP, as used by Jamal-Hanjani et al., exhaustively enumerates all clone trees, up to ten subclones. As such, for three patients (CRUK0032, CRUK0062 and CRUK0065) with more than ten subclones, CITUP could not be run and the authors constructed trees manually. Note that for these three patients, SubMARine identified subMARs in less than 40 seconds. For each tree, CITUP infers a set of subclonal frequencies that are close to the input frequencies and for which the associated clone tree is valid. Trees are ranked based on how close the input and inferred frequencies are, as assessed using a likelihood function. This function is maximized when the input frequencies already support a valid clone tree. As such, for the 42 patients which did not require a noise buffer, CITUP should find the same trees as SubMARine, assuming that only the most likely trees were reported. However, for six of the 42 patients, Jamal-Hanjani et al. report more trees. None of these additional trees were valid with the unaltered frequencies (see S22 Fig for an example). In 29 of the 46 cases requiring a noise buffer, the subMAR perfectly matches the trees reported by Jamal-Hanjani et al. Of the remaining 17 cases, in 13 cases, the valid trees completing the subMAR are a subset of the reported ones and in four cases, there is no overlap between reported and completing trees; however, the trees differ only in up to three parent-child relationships.

## Prostate cancer dataset

Finally, we applied SubMARine to a multi-sample metastatic prostate cancer dataset of Gundem et al. [42]. This dataset contains SSMs (both SNVs and indels) and CNAs from primary tumors and metastases of ten patients derived through whole genome sequencing (WGS), with some SSMs being validated through targeted deep sequencing. The ten patients each had between two and ten WGS samples. After detecting mutations and identifying subclones via clustering using a Dirichlet process mixture model [2, 44], Gundem et al. collected some further deep, targeted sequencing data to confirm the presence of all subclones in their assigned samples. They used the WGS and targeted data to infer cancer cell fractions (CCFs) for each subclone in each sample.

Gundem et al. constructed a clone tree for each patient by manually applying the pigeon-hole principle [2], which is an inference rule implied by the sum constraint, to these *raw* CCFs. In all cases, they needed to alter the CCFs in order to find ones that were consistent with a valid clone tree, we call these new CCFs the *adapted* CCFs.

To test SubMARine, we applied it to these prostate cancer data in two ways. First, we applied SubMARine in basic mode to the raw and adapted CCFs to determine whether the manual reconstructions would be the same as those found via our automated procedure. Second, because the data also contains CNAs, we applied SubMARine in extended mode. However, to do so, we needed to use a method that assigns CNAs to subclones, as this was not done by Gundem et al.

Applying SubMARine to either the reported raw or adapted CCFs gives nearly exactly the same trees, with differences coming largely because SubMARine does not require each cancer to have a single clonal population. To apply SubMARine, we converted the CCFs into subclonal frequencies by multiplying the CCFs with the reported purity (Section V.5 in S1 Text). Doing so permits clone trees to have multiple independent cancerous populations descending from the germline (i. e., to be multi-tumors). Indeed, even computing CCFs in the first place assumes that a clonal population exists. SubMARine built the partial clone trees for each patient and each input data in less than 2 seconds on a single thread with less than one Gb of memory. There were five patients—A10, A12, A17, A21, A32—where the clone trees recovered by SubMARine exactly matched those reported by Gundem et al., regardless of whether the subclonal frequencies were based on the raw or on the adapted CCFs. When based only on the raw CCFs, SubMARine also recovered the same clone tree for patient A31, and when based only on the adapted CCFs for patient A34, including uncertain parentage for one subclone. For the clone trees of the remaining patients (S23 Fig and S3 Table), some of the differences resulted because, unlike Gundem et al., SubMARine permits multi-tumors. In two cases, A22 and A29, this results in additional uncertainty in the clone tree. In one case, A24, SubMARine finds a multi-tumor solution that is a better fit to the raw and the adapted CCF data than the Gundem et al. solution. In another case, A31 based on the adapted CCFs, SubMARine finds uncertainty in the clone tree that was missed by the manual procedure used in Gundem et al. Finally, in only one case, A34 based on the raw CCFs, the adaption of the CCFs by Gundem et al. changes the structure of the best fitting clone tree. Thus, in general, we conclude that SubMARine could have been applied to the raw CCFs and completely replaced the manual procedure used by Gundem et al. This would have brought nearly the exact same results except that it would have made fewer minor errors in the construction of the clone trees.

To use SubMARine in extended mode, we needed subclonal assignments for the CNAs. Because Gundem et al. do not provide this information, we had to redefine the subclones and applied PhyloWGS [10] to the CNAs and the subset of the SSMs from Gundem et al. that were publicly available (Section V.6 in S1 Text). These SSMs constitute only those present in the

coding regions. Note further that Gundem et al. attempt to correct CCFs of SSMs for CNAs, however, their correction assumes that all cancer cells with an SSM have the same number of mutated alleles [44], which is not necessarily the case. Also, note that most CNA callers, including the one used by Gundem et al., are inaccurate when more than one CNA affects a region. As such, we used the PhyloWGS parsing mode that excludes SSMs in these multiple CNA regions, and only attempted clone tree reconstruction for those patients for whom less than half of their SSMs got excluded: A10, A12, A17, A24 and A34. For all the reason we mention above, the redefined subclones were not a perfect match to the Gundem et al. subclones. Hence, for the reported PhyloWGS tree with the highest likelihood we chose for each patient, we computed a mapping based on SNV assignment and subclonal frequencies between the subclones defined by PhyloWGS and those reported by Gundem et al. (Section V.6 in S1 Text and S4 Table). Because PhyloWGS does not phase SSMs in regions of copy number change, we computed the phasing with the highest likelihood in order to derive CNA influence on SSMs, which is needed for SubMARine.

SubMARine built the subMAR for each patient in less than 7 seconds on a single thread with less than one Gb of memory. Since we use the subclonal frequencies of the inferred clone trees from PhyloWGS as input, no noise buffer was necessary. Despite all of the differences in subclone construction between PhyloWGS and Gundem et al., our constructed partial clone trees largely match the ones reported by Gundem et al. (compare S24 Fig with Fig 2 of Gundem et al., and S3 Table). However, generally speaking, the PhyloWGS-output-based clone trees had more uncertain ancestor-descendant relationships; expect for patient A17, for which all relationships were defined. For patient A17, we could also match all, and for patient A34, all but one of PhyloWGS' subclones to the ones reported by Gundem et al. For these two patients, our partial clone trees resemble the reported ones best. Nevertheless, for all patients, the defined relationships between matched subclones never contradict the relationships reported by Gundem et al., however, SubMARine identifies new uncertain relationships unreported in Gundem et al. So, in conclusion, for the Gundem et al. dataset, adding CNAs directly to the clone trees did not change the reconstructions significantly and in addition permits to infer the evolutionary relationships between CNAs and SSMs. In the future, with more precise subclonal CNA reconstruction methods, these subclonal CNAs can be used by SubMARine to infer more detailed clone trees.

## Conclusion and discussion

Here we have introduced SubMARine, a polynomial-time algorithm that computes the *subMAR*, a partial clone tree that is a simple, partial solution to the NP-complete problem of finding a valid clone tree for a subclonal frequency matrix $\phi$. Despite that the subMAR is only an approximation, in all cases, when there is only a single clone tree solution, assuming precisely measured subclonal frequencies, SubMARine identifies it. Indeed, the subMAR only fails to capture the vast majority of the non-trivial ancestral relationships in the MAR when the reconstruction problem is severely under-constrained by the input data; and often these cases can be diagnosed by examining the subMAR. Notably, SubMARine also solves a potentially much more difficult extension of the basic clone tree reconstruction problem that includes subclonal CNAs (see also [45]). Furthermore, SubMARine permits the addition of user-defined ancestral constraints and SSM phasing, which could come from single cell or long read sequencing data. Additionally, we introduced a noise-buffered version of SubMARine to deal with imprecise subclonal frequencies. This version identifies the minimum uniform deviation from the input subclonal frequencies in order to prevent violations of the generalized sum rule and thus permits a valid partial clone tree for an input dataset. However, the uniform noise buffer can, in

some circumstances, generate highly uncertain subMARs when only a single (or small number) of input frequencies cause violations. To combat this issue, SubMARine attempts to identify a set of subclone- and sample-specific noise buffers in polynomial time. We are unable to provide guarantees that a given subMAR contains a valid, complete clone tree, nor can we guarantee that a uniform noise buffer can be made into a subclone- and sample-specific set of buffers in polynomial-time. As such, we introduce a depth-first search algorithm that is guaranteed to find the MAR with subclone- and sample-specific noise buffers if a valid, complete clone tree exists. However, among the reconstruction problems we considered, the combinatorial search is rarely needed for real-world problems or for well-specified simulated problems (i.e., those with few clone tree solutions).

Assuming precisely measured subclonal frequencies, SubMARine was able to construct the subMAR for nearly half of the TRACERx data where subclones were defined by mutation clustering. For the rest of the data, SubMARine could construct the subMAR using subclone- and sample-specific noise buffer sets. The noise-buffered version of SubMARine still requires an ordering of the subclones to initialize; the computation of this ordering does not consider the noise buffer and may be the source of differences between the solution sets reported by SubMARine and by CITUP on the TRACERx data.

When applying SubMARine on subclonal frequencies based on the adapted and raw CCFs of the prostate dataset by Gundem et al., we were able to identify some minor errors made in the previously expert-driven clone tree reconstructions. We showed that the partial clone trees based on the raw CCFs are similar to the ones based on the adapted CCFs, hence SubMARine applied to the raw CCFs would have been able to replace the manual procedure used by Gundem et al.

The partial clone tree introduced in this work is a particularly useful summary in domains, e. g. cancer therapy, where false positive claims on the evolutionary history of a tumor can have drastic consequences. Here, a conservative assessment of uncertainty is far superior to a random or representative single clone tree solution. Thus, an important use of SubMARine is generalizing a single clone tree—produced, e. g. through Monte Carlo sampling—to the set of valid and equivalent clone trees. As shown with the PhyloWGS clone trees produced with the Gundem et al. data, given a clone tree, one can easily derive the subclonal frequency matrix $\phi$; as well as defining the impact matrix $\mathcal{M}$ when SSM and CNA assignments are given. SubMARine can then identify the equivalence class of trees with equally good fits, thereby enhancing methods that give single or sampled solutions to a reconstruction problem.

Uncertainty in the actual values of the subclonal frequencies can give rise to incorrect reconstructions; SubMARine can be extended to address this challenge. Currently, when using noise buffers, SubMARine attempts to find the minimal deviation from the input subclonal frequencies necessary to find a valid partial clone tree. However, this could lead to overconfidence if there are other similar frequency deviations which give rise to different clone tree solution sets; this can be addressed using SubMARine in at least three ways. One solution would be to simply increase the size of the noise buffer (or buffer set) returned by SubMARine, and rerun it with higher-than-necessary buffers. One could also sample small amounts of noise and add these directly to the input subclonal frequencies; and combine subMARs resulting from multiple samples into a single one. Uncertainty between clone tree reconstructions can also be caused by differences in the clustering of mutations into subclones. As such, SubMARine can be used to enhance mutation clustering (or clone tree reconstruction) algorithms that output multiple solutions by applying SubMARine separately to each solution and merging the subMAR as above—however, this requires identifying a mapping between subclones. These approaches could thus be used to permit SubMARine to characterize the set of clone trees with nearly equivalent data fits; this may be especially important for datasets with low

purity or low sequencing depth where the input subclonal frequencies are likely to deviate the most from the actual ones.

There are a number of potential further extensions of this work. It may be possible to define the impact matrix $\mathcal{M}$ without a full subclonal reconstruction by adapting some of the pairwise comparisons technique developed in [45]. Indeed, it is possible to infer $\mathcal{M}$ directly for subclonal CNAs that are clonal in some, but not all, samples.

A potential drawback of SubMARine is the monotonicity restriction on the subclonal CNAs; note that this constraint is both more and less limiting than the infinite allele assumption previously applied to subclonal CNAs [45]. In particular, it effectively rules out incorporating clonal whole genome duplications (WGD) that appear in many cancers. It may be possible to extend SubMARine to incorporate clonal WGDs by expanding the number of potential phases for an SSMs.

There are a number of unanswered theoretical questions raised by this work. First, it is unclear what the hardness of the MAR reconstruction problem is. Because a MAR exists only if there is at least one valid clone tree solution, it seems likely that MAR reconstruction is at least as hard as the problem of finding a single clone tree solution. However, it is not clear whether this hardness changes under the assumption that a valid clone tree exists. Neither of these two questions are addressed by SubMARine. Also SubMARine approximates the MAR but provides no guarantees about its approximation factor. It would be useful to provide such guarantees, if they exist.

Finally, SubMARine could also be viewed as an extension of methods that perform haplotyping via perfect phylogeny [46, 47]. In quadratic time, these methods solve a special case of the basic clone tree reconstruction problem, in which all elements of the subclonal frequency matrix $\phi$ are either 0, 0.5, 1. Furthermore, they provide a complete, polynomial-space summary of all valid clone trees. Their summary methods could be generalized and applied to the possible parent matrix $\tau$ produced by Subpoplar.

## Supporting information

**S1 Fig. Example of a clone tree with three subclones and the germline.** Subclonal frequencies are indicated with $\phi_0, \ldots, \phi_3$; assuming that there are two samples given, their values could be $\phi_0 = (1, 1)$, $\phi_1 = (0.9, 0.8)$, $\phi_2 = (0.5, 0.3)$, and $\phi_3 = (0.4, 0.35)$. Edges between subclones indicate ancestral relationships, with the germline being an ancestor of all subclones and subclone 1 being the ancestor of subclones 2 and 3. Colorful bars indicate alleles of different segments; here, the two alleles of two segments are shown, with segment 1 having the dark gray and the light blue alleles, and segment 2 having the light gray and dark blue alleles. Two SSMs are assigned to subclone 1, one to the blue allele of segment 1 and one to the gray allele of segment 2. The SSMs are inherited by the descendants of subclone 1. Furthermore, two CNAs are assigned to the subclones, shown as copy number changes. One copy number duplication of the gray allele of segment 2 is assigned to subclone 2, duplicating the SSM lying on it. One copy number loss of the blue allele of segment 1 is assigned to subclone 3, deleting with it the SSM of this segment. (PDF)

**S2 Fig. Partial clone tree where one completing clone tree is not a solution to the basic clone tree reconstruction problem $t$.** Given $t$ with subclonal frequency matrix $\phi = (1, 0.7, 0.3, 0.2)^T$, this partial clone tree is its MAR. Six clone trees complete the MAR, however, only five of them are valid. The clone tree in which the germline is a parent of subclones 2 and 3 does not satisfy the sum constraint and hence is not a solution to $t$. (PDF)

**S3 Fig. Valid partial clone tree without a valid completion.** Example of a valid partial clone tree given the subclonal frequency matrix $\phi = ((1.0, 1.0), (0.6, 0.6), (0.4, 0.4), (0.39, 0.37), (0.38, 0.38), (0.37, 0.39))^T$. Subclones 1 and 2 are definite children of the germline. Subclones 1 and 2 do not have definite children because their ancestral relationships to subclones 3, 4 and 5 are undefined. In a completion without undefined relationships, either subclone 1 or 2 would have to have two definite children. However, given the frequencies in $\phi$, subclones 1 and 2 can have only one definite child without violating the generalized sum constraint. Thus, there exists no valid full completion of this valid partial clone tree.
(PDF)

**S4 Fig. Overview of SubMARine in basic mode.** The basic version of SubMARine takes the subclonal frequency matrix $\phi$ as input to build the ancestry matrix $Z$. In a preprocessing phase, the germline rule is introduced by setting $Z(0, k) = 1$ for all $k > 0$. Also, all trivial relationships are set to 0 ($Z(k, k') = 0$ for $k' \leq k$) as a consequence of the generalized sum constraint. Then, the main phase starts by using the crossing rule (Eq (9), Section III.1 in S1 Text), which also follows from the generalized sum constraint. The generalized sum rule itself and the partial tree rule are propagated by using Subpoplar until the ancestry matrix converged and no more relationships can be defined. Then, SubMARine outputs the ancestry matrix $Z$ together with the possible parent matrix $\tau$, created by Subpoplar. When the user defines additional constraints on $Z$, these are also input to SubMARine. They are applied after the preprocessing phase, followed by a propagation of the partial tree rule. This rule is also propagated now when using the crossing rule. The reason is that with the entries set by the user, $Z$ can contain 1's in other positions than the first row, possibly requiring updates of undefined relationships. Without user-defined constraints on $Z$, 1's in other rows can be set only in Subpoplar, hence the partial tree rule needs to be applied only at that stage. When the user provides clonal CNAs and SSMs as input, the lost allele constraint is checked before starting the preprocessing phase. Whenever a constraint cannot be satisfied, SubMARine terminates and indicates which subclonal relationship led to the conflict.
(PDF)

**S5 Fig. Overview of SubMARine in extended mode.** The extended version of SubMARine takes the subclonal frequency matrix $\phi$, CNAs as copy number changes in the matrices $\Delta C_A$ and $\Delta C_B$, assigned to subclones, segments and parental alleles in the vectors $\lambda_c$, $\sigma_c$ and $\pi_c$, SSMs assigned to segments and subclones in the vectors $\sigma_s$ and $\lambda_s$, and the impact matrix $\mathcal{M}$ as input to build the ancestry matrix Z and the SSM phasing vector $\pi_s$. At first, the monotonicity restriction is checked to hold on the CNAs. Then, in the preprocessing phase, the germline rule is introduced and trivial relationships ($Z(k, k') = 0$ for $k' \leq k$) are set. Afterwards, SubMARine starts the main phase, ensuring that the partial tree rule is applied each time a relationship is updated. First, the equivalence rule based on Eq (13) in Section IV.3 in S1 Text is propagated, leading to 1's in $Z$, together with those equivalence and lost allele rules that update SSM phasing. Second those equivalence and lost allele rules that lead to 0's in $Z$ and the crossing rule are used. Third, the general sum rule is propagated with Subpoplar, which also applies the equivalence, lost allele and partial tree rules whenever necessary. The method converges, when no more subclonal relationships and SSM phases can be updated. The output consists of the ancestry matrix $Z$, the SSM phasing vector $\pi_s$ and the possible parent matrix $\tau$, created by Subpoplar. The user can also define additional constraints on $Z$ and on $\pi_s$. Both types of constraints are applied after the preprocessing step and before the main phase starts. When user-constraints on $Z$ are set, the partial tree rule is already propagated before the main phase.

Whenever a constraint cannot be satisfied, SubMARine terminates and indicates what led to the conflict.
(PDF)

**S6 Fig. Proportion of subclones with uncertain parentage for (A) dataset without CNAs and (B) dataset with CNAs.** A subclone has uncertain parentage when it has multiple possible parents in the possible parent matrix $\tau$. Line shows mean and gray area standard deviation.
(PDF)

**S7 Fig. Proportion of subclones with uncertain parentage for dataset without CNAs containing (A) 5, (B) 20 and (C) 50 subclones.** A subclone has uncertain parentage when it has multiple possible parents in the possible parent matrix $\tau$. Line shows mean and gray area standard deviation.
(PDF)

**S8 Fig. Proportion of subclones with uncertain parentage for dataset with CNAs containing 20 subclones and different numbers of CNA events and segments.** (A)–(C) 10 segments, (D)–(F) 20 segments, (G)–(I) 40 segments, (A), (D), (G) 10 CNAs, (B), (E), (H) 20 CNAs, (C), (F), (I) 40 CNAs. A subclone has uncertain parentage when it has multiple possible parents in the possible parent matrix $\tau$. Line shows mean and gray area standard deviation.
(PDF)

**S9 Fig. Recall of defined ancestral relationships for datasets without and with CNAs and 20 subclones.** The recall is computed based on the non-trivial ancestral relationships. The p-value is computed with a Mann-Withney U test. The left column contains 193 data points and the right 1775.
(PDF)

**S10 Fig. Runtimes of the depth-first search (DFS) to enumerate all valid (and equivalent) clone trees completing a subMAR, sorted by the number of undefined ancestral relationships in the subMARs.** We terminated searches exceeding a maximal runtime of 120 h. We used two versions of the DFS to enumerate clone trees for different subMARs for the dataset without CNAs. The first version is a naïve, recursive one and the second version is an improved, iterative and also faster one, which we provide with SubMARine. Hence, if using the second version to enumerate the clone trees of all subMARs, the overall runtime could be improved. Note that for all subMARs on which the search did not termindate in 120 h, as well as for all subMARs of the dataset with CNAs, we already used the faster version.
(PDF)

**S11 Fig. Distribution among the different noise buffer statutes.** There are three different noise buffer statuses: Either no noise buffer is needed to build a subMAR, the subclone- and sample specific buffer set can be found in polynomial time, or the uniform buffer is used. The datasets with an effective read depth of 300 and 3000 contain 1200 subMARs, all others 600.
(PDF)

**S12 Fig. Maximum values in the noise buffer sets for different effective read depths.** Values for subMARs that could be built without a noise buffer are included as 0.
(PDF)

**S13 Fig. Proportion of subclones with uncertain parentage for dataset with noise and different effective read depths.** A subclone has uncertain parentage when it has multiple possible parents in the possible parent matrix $\tau$. Line shows mean. The lowest line for an effective read

depth of infinity shows the mean uncertain parentage of the corresponding noise-free simulated data.
(PDF)

**S14 Fig. Proportion of subclones with uncertain parentage for dataset with noise, 5 subclones and different effective read depths.** A subclone has uncertain parentage when it has multiple possible parents in the possible parent matrix $\tau$. Blue line shows mean and blue area standard deviation of uncertain parentage on noisy data. Orange line shows mean of corresponding noise-free data.
(PDF)

**S15 Fig. Proportion of subclones with uncertain parentage for dataset with noise, 20 subclones and different effective read depths.** A subclone has uncertain parentage when it has multiple possible parents in the possible parent matrix $\tau$. Blue line shows mean and blue area standard deviation of uncertain parentage on noisy data. Orange line shows mean of corresponding noise-free data.
(PDF)

**S16 Fig. Proportion of subclones with uncertain parentage for dataset with noise, 50 subclones and different effective read depths.** A subclone has uncertain parentage when it has multiple possible parents in the possible parent matrix $\tau$. Blue line shows mean and blue area standard deviation of uncertain parentage on noisy data. Orange line shows mean of corresponding noise-free data.
(PDF)

**S17 Fig. False positive errors of (A) falsely defining a relationship in the subMAR that is undefined in the noise-free MAR and (B) differently defining a relationship in the sub-MAR.**
(PDF)

**S18 Fig. (A), (D), (E) Recall and (B), (D), (F) false positive error of differently defining a relationship in the subMAR for (A), (B) 5, (C), (D) 20, and (E), (F) 50 subclones.** Orange thick bars show the mean recall and percentage of error of the noise-containing data, horizontal orange lines in (A), (C), (E) show the mean recall of the noise-free data. Note that the recall of the noise-free data was calculated by considering only the entries of the upper right triangle of the ancestry matrix $Z$, while for the noise-containing datasets also the entries of the lower left triangle were considered.
(PDF)

**S19 Fig. SubMARs for five patients from the TRACERx cohort.** Shown are the subMARs that contain undefined ancestral relationships. They are identical to their MARs. Subclonal indices are taken from the TRACERx mutation clusters.
(PDF)

**S20 Fig. Maximum values in the minimum noise buffer sets for 46 patients of the TRACERx cohort.**
(PDF)

**S21 Fig. (A) Subclonal frequencies and (B) partial clone tree built by SubMARine for patient CRUK0078 of the TRACERx study.** Subclonal indices are taken from the TRACERx mutation clusters. Both subclones 2 and 5 are children of subclone 1. However, they have a

subclonal frequency of 0.81 and 0.89, respectively, in sample 2. Hence, a noise buffer of 0.7 is necessary.
(PDF)

**S22 Fig. (A) Subclonal frequencies and (B) one clone tree built by CITUP in the TRACERx study for patient CRUK0095.** Subclonal indices are taken from the TRACERx mutation clusters. Given the shown subclonal frequencies and the clone tree, the sum constraint is not satisfied because $Z(2, 3) = 1$ although $\phi(2, 1) < \phi(3, 1)$. Hence, CITUP must have inferred other subclonal frequencies.
(PDF)

**S23 Fig. Partial clone trees built by SubMARine based on raw and/or adapted CCFs of Gundem et al.** The grey boxes show the parts of the partial clone trees that differ in Gundem et al., where for patient A22, the dark brown subclone with ID 23 is a child of the grey subclone with ID 2, for patient A24, the light blue subclone with ID 13 is a child of the orange subclone with ID 9, for patient A29, there is no uncertainty, the grey subclone with ID 5 is the parent of the dark pink subclone with ID 4 and the light blue subclone with ID 2, subclone 4 is the parent of the gold subclone with ID 3, and subclone 2 is the parent of the dark green subclone with ID 1, for patient A31, the light brown subclone with ID 7 is the child of the dark purple subclone with ID 3, and for patient A34, the darkbrown subclone with ID 11 could either be a child of the orange subclone with ID 18 or of the blue subclone with ID 20, and the pink subclone with ID 7 is a child of the light green subclone with ID 22. Note that the partial clone tree for patient A22 based on the raw CCFs does not contain uncertainty for subclone 8, other than in the tree reported by Gundem et al. Also note that the partial clone tree for patient A29 based on the raw CCFs does not allow the light blue subclone with ID 2 to be a parent of the dark green subclone with ID 1. Colors and subclonal IDs are taken from Gundem et al.
(PDF)

**S24 Fig. Partial clone trees built by SubMARine based on PhyloWGS' output.** For patient A10, subclone 13 is a possible child of all other subclones and the germline. Subclonal IDs are taken from the PhyloWGS trees. The colors are taken from Gundem et al. and show the mapping of PhyloWGS' subclones to the ones in Gundem et al. For patient A12 and A17, we mapped multiple PhyloWGS' subclones to the same subclone of Gundem et al. and in patient A17, we mapped two Gundem et al. subclones to subclone 2 of PhyloWGS. Subclones with a black stroke and grey filling could not be mapped to any Gundem et al. subclone. The letters below and next to the subclones show in which samples the subclones have a frequency higher than or equal to 0.1; ancestral subclones without an explicit labling combine the labeling of their descendants.
(PDF)

**S1 Table. Overview of inference rules derived from the lost allele and equivalence constraints.**
(PDF)

**S2 Table. SubMARine results on the TRACERx data.**
(XLSX)

**S3 Table. SubMARine results on the Gundem et al. data.**
(XLSX)

**S4 Table. Mapping between PhyloWGS and Gundem et al. subclones.**
(XLSX)

**S1 Algorithm. Functional description of the SubMARine algorithm in extended mode.**
(PDF)

**S1 Text. Supporting information on the lost allele constraint, on partial clone trees, on SubMARine in basic and extended mode, and on the results.**
(PDF)

**S1 Data. Excel spreadsheet containing the underlying numerical data for Figs 2–4, S6–S18, and S20 Fig.**
(XLSX)

## Acknowledgments

We thank Ben Raphael for constructive feedback and useful suggestions.

## Author Contributions

**Conceptualization:** Linda K. Sundermann, Jeff Wintersinger, Gunnar Rätsch, Jens Stoye, Quaid Morris.

**Data curation:** Linda K. Sundermann, Jeff Wintersinger.

**Methodology:** Linda K. Sundermann, Jeff Wintersinger, Quaid Morris.

**Project administration:** Linda K. Sundermann.

**Software:** Linda K. Sundermann.

**Supervision:** Gunnar Rätsch, Jens Stoye, Quaid Morris.

**Visualization:** Linda K. Sundermann, Jeff Wintersinger.

**Writing – original draft:** Linda K. Sundermann, Quaid Morris.

**Writing – review & editing:** Linda K. Sundermann, Jeff Wintersinger, Gunnar Rätsch, Jens Stoye, Quaid Morris.

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
