## [Decision Letter · Decision Letter 0]

16 Jun 2020

Dear Dr. Sundermann,

Thank you very much for submitting your manuscript "Reconstructing Tumor Evolutionary Histories and Clone Trees in Polynomial-time with SubMARine" for consideration at PLOS Computational Biology.

As with all papers reviewed by the journal, your manuscript was reviewed by members of the editorial board and by several independent reviewers. In light of the reviews (below this email), we would like to invite the resubmission of a significantly-revised version that takes into account the reviewers' comments.

We cannot make any decision about publication until we have seen the revised manuscript and your response to the reviewers' comments. Your revised manuscript is also likely to be sent to reviewers for further evaluation.

Sincerely,

Teresa M. Przytycka

Associate Editor

PLOS Computational Biology

Feilim Mac Gabhann

Editor-in-Chief

PLOS Computational Biology

Reviewer's Responses to Questions

**Comments to the Authors:**

Reviewer #1: This manuscript introduces SuBMARine, a method to infer a summary of phylogenetic trees that explain bulk sequencing data of tumors. Basically, given a solution space of trees the maximally-constrained ancestral reconstruction (MAR) designates all ordered pairs (i,j) of mutations as either ancestral (if i is ancestral to j in all trees), not ancestral (if i is not ancestral to j in all trees) or ambiguous (otherwise). Since exhaustive enumeration of the solution is intractable, the authors introduce a relaxation of the MAR, the subMAR,obtained directly from the frequency matrix. The subMAR (just like the MAR) is unique but may contain more ambiguous entries. In addition to the clean problem without CNAs, the authors consider a version of the problem with CNAs. I have two major comment and several minor comments.

Major:

1. Please provide a real data application of the extended SuBMARine algorithm.

While simulations are used to assess the performance of their algorithms in both problems settings, the TRACERx non-small-cell lung cancer data is used to assess only the basic version of the problem without CNAs. I would like to see a real application of the extended SuBMARine algorithm.

2. Dealing with uncertainty

Compared to the conference version, this manuscript contains an extension of the algorithm to account for uncertainty in the frequency matrix. It would be good to include more methodological details in the main text about this. Moreover, it would be good to assess the performance of SuBMARine using simulated data where one accounts for uncertainty in read counts.

Minor:

* Pseudo code:

- Line 1: As \\phi is a matrix, I would not write |\\phi| to indicate the number of rows. Simply define \\phi to be a K by N matrix.

- What is Equation 9? (this equation is also referred to several times in the main text).

* Author summary: "up o 50" => "up to 50"

* Line 140, 144, etc. What is t? Isn't the basic clone tree reconstruction problem defined by phi?

* Line 230. Elaborate on sorting of phi. Do you sort in ascending order of frequencies in first sample? How do you break ties?

Reviewer #2: The authors have significantly improved the manuscript. The methodology is clearly explained and supported by new figures and supplementary materials. The revised version of the paper covers some crucial topics more extensively, such as: 1) The authors have introduced a noise buffer to extend SubMARine to handling some noise in the estimates of subclonal frequency when performing reconstructions; 2) Full comparison of the results with CITUP method.

The changes introduced to the algorithm in this updated version address satisfactorily my concerns in my first round of review. In my opinion the methods and the manuscript have enhanced scientific quality addressing the challenges of clone tree reconstruction.

**Have all data underlying the figures and results presented in the manuscript been provided?**

Reviewer #1: Yes

Reviewer #2: Yes

PLOS authors have the option to publish the peer review history of their article (what does this mean?). If published, this will include your full peer review and any attached files.

Reviewer #1: No

Reviewer #2: Yes: Alexander Martinez-Fundichely
---

## [Decision Letter · Decision Letter 1]

22 Sep 2020

Dear Dr. Sundermann,

We are pleased to inform you that your manuscript 'Reconstructing Tumor Evolutionary Histories and Clone Trees in Polynomial-time with SubMARine' has been provisionally accepted for publication in PLOS Computational Biology.

Best regards,

Teresa M. Przytycka

Associate Editor

PLOS Computational Biology

Feilim Mac Gabhann

Editor-in-Chief

PLOS Computational Biology

Reviewer's Responses to Questions

**Comments to the Authors:**

Reviewer #1: My comments have been satisfactorily addressed.

**Have all data underlying the figures and results presented in the manuscript been provided?**

Reviewer #1: Yes

PLOS authors have the option to publish the peer review history of their article (what does this mean?). If published, this will include your full peer review and any attached files.

Reviewer #1: No

---

## [Editor Report · Acceptance letter]

11 Jan 2021

PCOMPBIOL-D-20-00692R1 

Reconstructing tumor evolutionary histories and clone trees in polynomial-time with SubMARine

Dear Dr Sundermann,

I am pleased to inform you that your manuscript has been formally accepted for publication in PLOS Computational Biology. Your manuscript is now with our production department and you will be notified of the publication date in due course.

With kind regards,

Jutka Oroszlan
